# Physiotherapists clinical reasoning to prescribe exercise for patients with chronic pain: A qualitative study research protocol

**Michael C. Kelly** [1]*, **Jenni Naisby** [1], **David J. Bell** [1,2]

1 Department of Sport, Exercise and Rehabilitation, Faculty of Health and Life Sciences, Northumbria University, Newcastle Upon Tyne, United Kingdom, 2 Northumbria Healthcare NHS Foundation Trust, Newcastle Upon Tyne, United Kingdom

☯ These authors contributed equally to this work.

* michael4.kelly@northumbria.ac.uk

## Abstract

### Background

Physiotherapists' play a key role in the management of chronic pain, and as part of the National Institute for Health and Care Excellence (NICE) guidelines, prescribe exercise to support patients with chronic pain. However, there is very limited evidence supporting physiotherapists on what type of exercise or dose of exercise should be prescribed. Physiotherapists' therefore have more onus on their ability to clinically reason how to prescribe exercise. At present, there is no research investigating how physiotherapists' working with patients that have chronic pain, clinically reason when prescribing exercise. This study proposes to investigate how physiotherapists experienced in pain management prescribe exercise, to understand what the key influences are on their reasoning, and how these impact on clinical practice.

### Methods

This will be a qualitative study, utilising semi-structured individual interviews. Participants will be Health and Care Professions Council registered physiotherapists, working predominantly with patients that have chronic pain. Recruitment will focus on physiotherapists working within the United Kingdom (UK). Up to twenty participants will be recruited. The study, including the interview guide, will be supported by a steering group consisting of academics and physiotherapists experienced in chronic pain. The data will be analysed using framework analysis.

### Results

The study will be reported using the COnsolidated criteria for REporting Qualitative research (COREQ) guidelines. The findings of the study will be disseminated through publication in a peer reviewed journal.

**Data Availability Statement:** No datasets were generated or analysed during the current study. All

relevant data from this study will be made available upon study completion.

**Funding:** The author(s) received no specific funding for this work.

**Competing interests:** The authors have declared that no competing interests exist.

## Conclusion

This study will provide novel insight into how physiotherapists experienced working with and managing chronic pain patients, prescribe exercise, and will gain new insight into clinical practice to help inform future research and education.

## Introduction

Chronic pain, also referred to as persistent pain, is a pain that lasts more than three months [1]. Although there is limited consensus, it is estimated that one-third or up to one-half of the population are affected by chronic pain [1]. A meta-analysis by Fayaz and colleagues [2] reported a prevalence of 35–51.3% within the United Kingdom (UK). Chronic pain is a leading cause of disability, which can affect work, relationships and self-esteem [3]. Patients with chronic pain have described the negative interference it has with physical functioning, professional life, family and relationships, social life, sleep and mood [4]. At present, chronic pain exerts an enormous burden from individual, societal and economic viewpoints [3, 5].

Management of chronic pain is complex, requiring a person-centred assessment [1] and multimodal approach [3]. The use of exercise and physical activity (PA) are two of the recommended approaches to non-pharmacological management of persistent pain [1]. Although the terms exercise and physical activity have been used interchangeably within the literature [6], there is a clear distinction between each term. Physical activity is defined as any bodily movement produced by skeletal muscles that results in energy expenditure [7], whereas exercise is a subset of physical activity, that is planned, structured and repetitive and aims to improve or maintain physical fitness [7]. As this study is focused on the prescription of exercise, the term exercise will be used throughout.

Physiotherapists' play a key role in pain management as part of a multidisciplinary team [8] and are in an ideal position to prescribe exercise and support PA, as they are able to support physical function, provide advice, explanation and encourage activity [9]. Additionally, physiotherapists' are able to assess risk versus safety in human movement and play a key role in reducing disability in chronic pain [10].

Although exercise and PA are key recommendations for the management of chronic pain and form part of the NICE [1] guidelines, there is little detail about how to put the recommendations into practice. There is an absence of evidence to guide physiotherapists' on the prescription of the type of exercise, and exercise dose, which is subject to considerable uncertainty [11, 12]. The evidence used to inform the NICE [1] guidelines was predominantly based on female patients with fibromyalgia syndrome (FMS) and patients with neck pain. The guidelines stated that limited evidence was available to compare different types of exercise, making no distinction in which exercise (cardiovascular, mind–body, strength, or a combination of approaches) would be preferential for patients. In their chronic pain update published in the Lancet, Cohen and colleagues [3] also recognised a lack of evidence guiding one type of exercise over another for chronic pain, while highlighting that exercise appears more beneficial for function as opposed to pain relief. A recent network meta-analysis [13], focusing on chronic low back pain, did conclude that Pilates, McKenzie therapy and functional restoration were more effective for reducing pain and functional limitation, compared to other exercise. However, the study did also report that other types of exercise, other than stretching, were more effective for reducing pain and improving functional limitations compared to minimal care, and a highlighted that a higher dose (measured with time) or adding co-interventions for most exercise treatments, was more effective than minimal care.

A recent meta-analysis [14], systematic review with a meta-analysis [15], and Cochrane Review [16] have highlighted that while evidence supports the use of exercise for chronic pain, there is virtually no knowledge of the appropriate dose [14], and there is a scarcity of evidence supporting physiotherapists' on the parameters of exercise prescription, and how they may change prescription, when faced with the predominance of nociplastic pain [15]. Although daily recommended exercise guidelines exist [17], it has been postulated that these may be an excessive dose for patients with chronic pain [14]. Whilst the NICE [1] guidelines are unable to direct physiotherapists' on the most effective type of exercise, they do state that patients would be more likely to exercise if they were provided with programmes that suited their lifestyle, physical ability and addressed their health needs. Individualising treatment is important [12], but doing so complicates the ability to establish definitive exercise prescription guidelines [11].

Despite the lack of evidence and recommendations for physiotherapists' to prescribe exercise, daily, physiotherapists' are required to prescribe exercise and support physical activity for patients with chronic pain. Clinical reasoning is a key component for physiotherapists', defined as a "context dependent way of thinking and decision making in professional practice to guide practice actions" [18] and a foundation of physiotherapy practice. Although studies have investigated how physiotherapists' clinically reason in the assessment of pain [19] or how ready they are to manage chronic pain [20], there appears to be no research investigating how physiotherapists' clinically reason to prescribe exercise for patients with chronic pain. Ultimately, the guidance to prescribe exercise is variable and generic for patients with chronic pain, the result of which is a reliance on individual physiotherapists' preferences and abilities to prescribe exercise [15].

Considering the very limited guidance to prescribe exercise for pain, it is important to understand how physiotherapists' who are experienced in pain management clinically reason to prescribe exercise and support rehabilitation. Rehabilitation is based upon social interaction and dependent upon attitudes, thoughts, and motivations [21] and qualitative research is well placed to help gain insight into what is important for those involved in the research [21]. Therefore, investigating how physiotherapists' experienced in pain management prescribe exercise, will help understand what the key influences are on their reasoning, and how these impact on clinical practice. Doing so, will help guide other physiotherapists or health care professionals to consider what may support their own clinical reasoning, to help prescribe exercise for patients with chronic pain. This work will highlight key opportunities for physiotherapists, physiotherapy education and further research to understand the key aspects of clinical reasoning when prescribing exercise for patients with chronic pain, but also provide a starting point to improve how exercise is prescribed for patients with chronic pain.

## Methods

### Study design and type

In this study, following a generic qualitative approach [22], semi-structured individual interviews will be used to gather qualitative data, with the aim of investigating the clinical reasoning of physiotherapists' prescribing exercise for patients with chronic pain. The interview guide (see S1 File) has been developing through a scope of the literature and the research team experiences. The research teams experiences were deemed important, due to limited extant literature. The research team piloted the questions using a steering group of physiotherapists' and academics that were experienced in working with patients that have chronic pain. This steering group were consulted regarding the interview content guide, as well as the research concept. The research team are all musculoskeletal physiotherapists from different career backgrounds.

There is one clinical pain specialist working within the NHS (DB) and two Assistant Professor physiotherapists, experienced in research in the areas of pain (JN) and exercise (MK). The guide includes topics such as engagement with the guidelines, influences on reasoning (such as prior experience, education, patient viewpoints/expectations), influence of type of pain/pain mechanisms, service agreements and therapist experience.

## Study place and population

Recruitment will begin in summer 2023, lasting up to one year. A purposive sample of Health and Care Profession Council (HCPC) registered physiotherapists, working predominantly or exclusively with chronic pain patients (such as within pain services or pain clinics) that present with chronic primary pain (such as, but not exclusively fibromyalgia syndrome) [23] or chronic secondary pain (such as, but not exclusively osteoarthritis) [24] will be recruited.

Recruitment will not take place directly within health services (such as the National Health Service), but through adverts circulated via social media, the Physiotherapy Pain Association, the Physiotherapy Research Society as well as using a snowballing approach using clinical experts within physiotherapy networks to support recruitment. Potential participants will contact the principal researcher via email, and be provided (via email) with an electronic participant information sheet and consent form. Participants will provide consent by emailing a completed consent form to the principal researcher. Following consent, a date and time will be agreed for the interview to take place.

## Sample size

Providing an a-priori sample size within qualitative research has been subject to criticism, based upon the premise that it is illogical to suggest a specific number of participants based upon a subject or topic with which the key themes cannot be identified in advance, or create understanding about what is as of yet known [25].

This study will recruit participants using the concept of information power [26], whereby the more information a sample holds, relevant for the study, the lower the amount of participants are required. This approach allows researchers to approximate the number of participants required for analysis. Appraisal of information power should be repeated as stepwise approach [26] and will be following each interview in this study.

Although using information power to guide sample size provides guidance, it is not without criticism and as with all qualitative research, justifying a sample size is a genuine challenge [25]. Therefore, as opposed to stipulating a specific number of participants, this study will impose an upper limit of 20 participants, following the guidance of Sim [25]. This guidance, which also mirrors that of Malterud [26], recommends ongoing interpretation of the data by the research team, using an inductive approach, with a focused research agenda, to increase the potential for saturation [27], or in this case information power, with the aim to understand how physiotherapists' experienced in pain management prescribe exercise, and understand what the key influences are on their reasoning, and how this impacts on clinical practice.

## Data collection

All interviews will be conducted by the first author (MK)The study will take place face to face with participants on the university grounds, or via teleconferencing. All interviews will take place in a private space. All interviews will be audio recorded using two digital voice recorders. If interviews are undertaken via teleconferencing, no screen footage will be recorded, only an audio recording. The interviews will last up to two hours to allow exploration on the participants' clinical reasoning. Field notes made during the interviews may also be taken, to record

key insight or aspects of the interviewer's experience. Prompting and probing questions will be used to support the interview guide. Following each interview, the interview questions will be reviewed, adapted or refined by MK and JN, as part of a reflexive approach.

The age, level of entry qualification, highest level of qualification, number of years qualified and number of years working with chronic pain will be collected. This will enhance the transferability of study findings. No other data will be collected, other than the contents of the interviews, which will be audio recorded.

## Data management and analysis

Audio recordings will be transcribed, and participants will be allocated a numeric code at the beginning of the study, which will be used to identify any data which they provide within the study. None of the participants' personal details will be associated with any participant data. Informed consent forms containing both name and code will be stored separately from the recorded data in the study. Framework analysis [28] will be used to analyse the data. Framework analysis, consisting of seven stages (Transcription, familiarisation, coding, developing and analytical framework, applying the analytical framework, charting data into framework matrix and interpretation), is a form of thematic analysis, and thematic analysis is not tied to a specific discipline or construct [29], which helps to provide theoretical freedom and flexibility, while providing rich, detailed and complex accounts of data [30]. The analysis will be conducted by members of the research team independently, therefore providing the benefit of multiple coding, independent verification of analysis [31] and code refinement [32]. Trustworthiness will also be enhanced using guidance by Shenton [33], which includes the use of frequent debriefing sessions between the research team, peer scrutiny, and the use of reflexive diaries as the research develops.

## Ethical considerations

The study has been approved by Northumbria University Ethics Committee (11th July 2022, Ref 51228). Following completion of the study, participants will receive an electronic participant de-brief, informing them of the lead researcher contact details, ethics officer details and information about what the information from the study will be used for.

## Dissemination of results

The study will be reported using the COnsolidated criteria for REporting Qualitative research (COREQ) guidelines [34]. The findings of the study will be disseminated by sharing at a national conference and publication in a peer reviewed journal. In addition, a summary will be shared with professional networks who have supported recruitment.

## Discussion

To the best of our knowledge, this will be the first study seeking to understand the clinical reasoning processes of physiotherapists' when prescribing exercise or PA for patients with chronic pain. There is a paucity of evidence guiding physiotherapists' about the type or dose of exercise for patients with chronic pain. Therefore, investigating how physiotherapists' experienced working with and managing chronic pain patients, prescribe exercise or PA, will gain new insight into clinical practice. In order to improve research within healthcare, it should be designed, disseminated and implemented with stakeholder input, such as clinical staff and patients [35]. Therefore, using the support of a steering group of physiotherapists' and academics, we will seek to understand what the key influences are on the physiotherapists' clinical

reasoning processes. This will include how much importance is placed on exercise and, how they may gain a baseline for exercise to start with, and how they progress and monitor patients. This study will also consider the range of influences on their clinical reasoning, including the patient presentation, the patients' views and goals, the current evidence base, as well as the physiotherapists' backgrounds and experiences of managing patients with chronic pain. This research will also consider what concerns physiotherapists' have regarding exercise prescription and highlight what questions they may pose when looking forward to improving their understanding of exercise prescription for their patients.

Understanding the reasoning processes of physiotherapists' will help provide new insight about how patients with chronic pain are supported using exercise. Additionally, this study will help inform future research, other health care professionals involved with patients that have chronic pain, and undergraduate education, for this important, but poorly understood topic.

## Study limitations

This study will investigate the views of HCPC registered physiotherapists' only, therefore specific contextual influences (for example, individual service provision) will be unique to physiotherapists' working within the United Kingdom. This study will also be specific to physiotherapists' who work predominantly with patients that have chronic pain. Therefore, this study will not reflect the experiences or reasoning of other physiotherapists' that may encounter patients with chronic pain, as part of a typical caseload. Although the geographical reach of the research may be limited, and specific to a small proportion of physiotherapists that specialise in chronic pain, and therefore limiting transferability, this study will provide important contextual insight into how physiotherapists' working in the UK, prescribe exercise for patients with chronic pain. To enhance transferability, it will be ensured that detailed contextual information will be provided about the sample.

Understanding context within research and clinical practice is important. Patient preferences and concerns are shaped by contextual factors which impact on applicability of clinical guidelines [36], while studies that do not consider context, risk being unable to articulate the "how" and the "why" [37]. Due to the paucity of evidence guiding physiotherapists' in exercise prescription for chronic pain, developing research that can capture the contextual factors that impact on physiotherapists' clinical reasoning is important to furthering our understanding of this topic.

## Supporting information

**S1 File. This is the semi structured interview guide to investigate physiotherapists' clinical reasoning to prescribe exercise for patients with chronic pain.**
(DOCX)

## Author Contributions

**Conceptualization:** Michael C. Kelly, David J. Bell.

**Methodology:** Michael C. Kelly, Jenni Naisby, David J. Bell.

**Writing – original draft:** Michael C. Kelly.

**Writing – review & editing:** Jenni Naisby, David J. Bell.

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
