## [Decision Letter · Decision Letter 0]

14 Apr 2023

PONE-D-23-00756Physiotherapists clinical reasoning to prescribe exercise for patients with persistent pain: a qualitative study research protocol.PLOS ONE

Dear Dr. Kelly,

Thank you for submitting your manuscript to PLOS ONE. After careful consideration, we feel that it has merit but does not fully meet PLOS ONE’s publication criteria as it currently stands. Therefore, we invite you to submit a revised version of the manuscript that addresses the points raised during the review process.

The reviewers recommended major revisions. The comments provided by the reviewers are included at the end of this email. Please address each one before submitting your revision.

We look forward to receiving your revised manuscript.

Kind regards,

Dr Claudio Di Lorito

Academic Editor

PLOS ONE

Journal Requirements:

Reviewers' comments:

Reviewer's Responses to Questions

**Comments to the Author**

1. Does the manuscript provide a valid rationale for the proposed study, with clearly identified and justified research questions?

Reviewer #1: Yes

2. Is the protocol technically sound and planned in a manner that will lead to a meaningful outcome and allow testing the stated hypotheses?

Reviewer #1: Yes

3. Is the methodology feasible and described in sufficient detail to allow the work to be replicable?

Reviewer #1: No

4. Have the authors described where all data underlying the findings will be made available when the study is complete?

Reviewer #1: Yes

5. Is the manuscript presented in an intelligible fashion and written in standard English?

Reviewer #1: Yes

6. Review Comments to the Author

You may also provide optional suggestions and comments to authors that they might find helpful in planning their study.

Reviewer 1 comments:

INTRODUCTION

• You are using exercise and physical activity interchangeably. I suggest you stick to exercise and not physical activity since physical activity cannot be prescribed (line 150, 155). It is exercise that can be prescribed.

METHODS

A lot of information has been mixed up in the methods section. Authors are providing information that is not necessary for a publication. For example, some details (lines 215 to 252) can be summarized in a single sentence and then just referenced. Authors are writing the methodology section as if this is a thesis, yet this is a manuscript for publication. I suggest you cut down any unnecessary details in the methods section and write the methodology section concisely with the following subheadings:

• Study design and type

• Study place

• Study population

• Sample size/ sampling

• Data collection

• Data management and analysis

• Ethical considerations

• Dissemination of results

Some further comments in the methods section are below:

Study design

• Information under study is largely describing the data collection process and the tools used for data collection. Authors need to indicate what study design will be used as this has not been described under the study design section.

• Some information in lines 181 to 187 is misplaced. This information should come the ethical consideration. In addition, this section (ethical considerations) is missing and needs to be included.

Study population

• Information on sample size has been misplaced (lines 253 to 263) under data collection. This information should be under a sub heading sample size which follows the study population sub section.

• Under sample size, authors should be specific on how participants will be sampled. What sampling frame will be used?

Data analysis

• Provide information on how data trustworthiness will be ensured.

DISCUSSION

Study limitations

• Since this is a protocol, indicate how you intend to mitigate the limitations that you foresee.

Reviewer 2 comments:

Thank you for submitting this protocol for review. It is certainly a useful piece of work to understand the clinical reasoning of clinicians and something that has been poorly explored previously. I have outlined a few questions and amendments to strengthen the protocol

Abstract

Background – uses NICE abbreviation, suggest no abbreviations on an abstract, spell out in full for international readers. Can we avoid using the word patients as many service users do not view themselves as patients, particularly in the outpatient setting

Methods – again using UK abbreviation Write out in full

Ethics statement – needs more detail. Please explain why NHS IRAS/HRA ethics will not be required. What do you mean consent will be gained in the written format. Please give further details. Has ethics already been obtained?

Data section (page 5). I am confused as you state that no data sets were produced or analysed as part of the study but this study is generating a new data set (qualitative data from interviews). Please clarify and amend

Full manuscript

Background

- You state that chronic pain and persistent pain are more or less the same. Why have you chosen to call it persistent pain throughout the manuscript as this could be confusing for your reader?

- Chronic/persistent pain is complex. Are you focusing on clinical reason in certain conditions/populations (i.e MSK related, organ related, multi-morbidity, fibromyalgia)? Clarifying this would help focus the mind of your readers.

- Please define exercise and physical activity (as they are two distinct, though related terms)

- Although you have focused on physiotherapists as the main prescribers of exercise, I think you could expand the final paragraph of this section (page 14) to consider how other professionals could learn from these findings. For example exercise physiologists etc

Method

- You state that the interview guide has been developing using literature. Can you elaborate? Have you completed a systematic/mapping review to identify literature? How can you be sure there isn’t any available evidence?

- Please confirm if consent will be written or electronic

- Robust qualitative research includes reflexive diaries. I note you have said that you will keep field notes but reflexive diaries are also needed and should be incorporated into the analytical framework

- I am unclear why you need two people to complete a semi-structured interview. Why would JN observe and make notes? This is unsual for this method so please explain the rationale. The bias of two interviewers to one participant could potentially introduce bias into the method through social power. This needs to be explained further

- If you are using data saturation to determine your sample size please define your working definition of saturation

- There is limited detail on the analysis section. I note that you are suing the framework method which has seven stages of analysis (linked to the Gale reference). The authors have not followed this nor provided enough detail on their analytical method. For this to be published the seven stages need to be outlined.

7. PLOS authors have the option to publish the peer review history of their article (what does this mean?). If published, this will include your full peer review and any attached files.

Reviewer #1: **Yes: **Dr. Enock Chisati

---

## [Author Response · Author response to Decision Letter 0]

18 May 2023

Dear reviewers, 

Thank you for taking the time to review our manuscript, and for providing clear and detailed feedback to help improve the submission. Below, we have provided a table with reviewer comments in one column and responses in another, to help provide a clear audit of the changes made. We have responded to each reviewers points separately, though some points made by the reviewers highlight the same issue (such as terminology relating to exercise and physical activity). 

Reviewer 1

INTRODUCTION

• You are using exercise and physical activity interchangeably. I suggest you stick to exercise and not physical activity since physical activity cannot be prescribed (line 150, 155). It is exercise that can be prescribed.

Response: Agreed, exercise will be utilised throughout. Reviewer 2 requested that exercise and physical activity should be defined, which has been added (lines 96-104) and is followed by why exercise will be used throughout the manuscript.

METHODS

A lot of information has been mixed up in the methods section. Authors are providing information that is not necessary for a publication. For example, some details (lines 215 to 252) can be summarized in a single sentence and then just referenced. Authors are writing the methodology section as if this is a thesis, yet this is a manuscript for publication. I suggest you cut down any unnecessary details in the methods section and write the methodology section concisely with the following subheadings:

• Study design and type

• Study place

• Study population

• Sample size/ sampling

• Data collection

• Data management and analysis

• Ethical considerations

• Dissemination of results

Response: An effort was made to be more succinct in reporting of the methods in lines 207-227. However, reviewer two commented that a definition was required relating to information power/saturation, therefore this has remained in the script. We have amended the script and used the titles and order of titles as suggested. However, we have combined study place/population as a sub heading, as each section in isolation would have limited detail. We have added a sub heading for ethical considerations (lines 263-268) and dissemination of results respectively (270-273) 

Some further comments in the methods section are below:

Study design

• Information under study is largely describing the data collection process and the tools used for data collection. Authors need to indicate what study design will be used as this has not been described under the study design section.

• Some information in lines 181 to 187 is misplaced. This information should come the ethical consideration. In addition, this section (ethical considerations) is missing and needs to be included

Response: We have clarified that this will be a qualitative study, following a generic qualitative approach as described by Kahlke (2014) on lines 172-175. Ethics information has been moved to the “ethical considerations” section (lines 263-268)

Study population

• Information on sample size has been misplaced (lines 253 to 263) under data collection. This information should be under a sub heading sample size which follows the study population sub section.

• Under sample size, authors should be specific on how participants will be sampled. What sampling frame will be used?

Response: We have moved the information on sample size to the appropriate sub heading of sample size (lines 206-227). As described in lines 190, a purposive sample will be used from the recruitment method outlined in lines 196-199. We felt that this approach was most appropriate, due to the nature of the physiotherapists linked to these associations.

Data analysis

• Provide information on how data trustworthiness will be ensured

Response: This has been added in lines 259-261

DISCUSSION

Study limitations

• Since this is a protocol, indicate how you intend to mitigate the limitations that you foresee.

Response: We have highlighted that the main limitation is the transferability of the study, due to the specific type of clinicians to be recruited, and highlighted how this could be mitigated in lines 313-314

Reviewer 2

Background – uses NICE abbreviation, suggest no abbreviations on an abstract, spell out in full for international readers.

Response: Agreed, this has been amended accordingly.

Can we avoid using the word patients as many service users do not view themselves as patients, particularly in the outpatient setting

Response: We have elected to keep the word patient, this is to keep aligned with the terminology used by National Institute for Health and Care Excellence (NICE), and the International Association for the Study of Pain (IASP).

Methods – again using UK abbreviation Write out in full

Response: Agreed, this has been amended accordingly.

Ethics statement – needs more detail. Please explain why NHS IRAS/HRA ethics will not be required. What do you mean consent will be gained in the written format. Please give further details. Has ethics already been obtained?

Response: Yes, ethics has been obtained. This was stated in lines 187-188 (in the original submission), detailing the ethics committee, reference number, and the date ethics was obtained. With the amendments to the manuscript, this is now on lines 263-268. Consent will be electronic, via email. This has been clarified in lines 200-204. As recruitment is not taking place directly within the NHS, but via social media, the Physiotherapy pain association and the Physiotherapy research society and can include participants that are not working in the NHS, NHS IRAS/HRA ethics is not required. We have highlighted in the manuscript that recruitment will not be directly within the NHS (lines 196-199)

Data section (page 5). I am confused as you state that no data sets were produced or analysed as part of the study but this study is generating a new data set (qualitative data from interviews). Please clarify and amend

Response: The data collection has not started. For clarity, we have amended the date of the recruitment to summer 2023 (line 190). We did not want to/have not started data collection, as we wanted this protocol paper to be accepted first as per the PLOS one guidelines.

Full manuscript

Background

You state that chronic pain and persistent pain are more or less the same. Why have you chosen to call it persistent pain throughout the manuscript as this could be confusing for your reader?

Response: We have amended this to chronic throughout the script, which includes the title

Chronic/persistent pain is complex. Are you focusing on clinical reason in certain conditions/populations (i.e MSK related, organ related, multi-morbidity, fibromyalgia)? Clarifying this would help focus the mind of your readers.

Response: This has been clarified in the participants section (lines 190-195) and uses IASP references to support

Please define exercise and physical activity (as they are two distinct, though related terms)

Response: We have defined both terms (lines 96-104), as physical activity is discussed in the NICE guidelines, but also highlight why exercise will be used throughout the manuscript

Although you have focused on physiotherapists as the main prescribers of exercise, I think you could expand the final paragraph of this section (page 14) to consider how other professionals could learn from these findings. For example exercise physiologists etc

Response: We have added to the final paragraph, but opened this up to health care professionals (line 298)

Method

You state that the interview guide has been developing using literature. Can you elaborate? Have you completed a systematic/mapping review to identify literature? How can you be sure there isn’t any available evidence?

Please confirm if consent will be written or electronic

Response: A systematic review/mapping review was not undertaken. However, the scope of the literature, as far as the research team was aware was used, due to the limited exploration of this subject. We had highlighted in the script that the scope of the literature as far as the research team was aware was utilised (lines 175-177) . Consent is electronic (lines 200-204)

Robust qualitative research includes reflexive diaries. I note you have said that you will keep field notes but reflexive diaries are also needed and should be incorporated into the analytical framework

Response: This has been added in lines 260-261

- I am unclear why you need two people to complete a semi-structured interview. Why would JN observe and make notes? This is unsual for this method so please explain the rationale. The bias of two interviewers to one participant could potentially introduce bias into the method through social power. This needs to be explained further

Response: We have removed JN from attending the interviews.

If you are using data saturation to determine your sample size please define your working definition of saturation.

Response: We are not using data saturation, but information power, which is covered in lines 211-216

There is limited detail on the analysis section. I note that you are suing the framework method which has seven stages of analysis (linked to the Gale reference). The authors have not followed this nor provided enough detail on their analytical method. For this to be published the seven stages need to be outlined.

Response: We have added the seven stages of framework analysis to this section (lines 251-253) but have not discussed each, as the study has not yet been conducted.

---

## [Decision Letter · Decision Letter 1]

7 Aug 2023

PONE-D-23-00756R1Physiotherapists clinical reasoning to prescribe exercise for patients with chronic pain: a qualitative study research protocol.PLOS ONE

Dear Dr. Kelly,

Thank you for submitting your manuscript to PLOS ONE. After careful consideration, we feel that it has merit but a few minor comments need addressing to fully meet PLOS ONE’s publication criteria as it currently stands. Therefore, we invite you to submit a revised version of the manuscript that addresses the points raised during the review process.

We look forward to receiving your revised manuscript.

Kind regards,

Claudio Di Lorito

Academic Editor

PLOS ONE

Journal Requirements:

Additional Editor Comments:

Dear Authors

Thank you for the revisions. Two reviewers have checked your re-submission and they are happy to proceed with publication if you addressed the minor comments included. Please carefully addressed these reminaing comments, so we can proceed with publication of your manuscript.

Best Wishes

Claudio Di Lorito

Reviewers' comments:

Reviewer's Responses to Questions

**Comments to the Author**

1. Does the manuscript provide a valid rationale for the proposed study, with clearly identified and justified research questions?

Reviewer #2: Yes

Reviewer #3: Yes

2. Is the protocol technically sound and planned in a manner that will lead to a meaningful outcome and allow testing the stated hypotheses?

Reviewer #2: Yes

Reviewer #3: Yes

3. Is the methodology feasible and described in sufficient detail to allow the work to be replicable?

Reviewer #2: Yes

Reviewer #3: Yes

4. Have the authors described where all data underlying the findings will be made available when the study is complete?

Reviewer #2: Yes

Reviewer #3: Yes

5. Is the manuscript presented in an intelligible fashion and written in standard English?

Reviewer #2: Yes

Reviewer #3: Yes

6. Review Comments to the Author

You may also provide optional suggestions and comments to authors that they might find helpful in planning their study.

Reviewer #2: I would like to congratulate the work done previously by the authors and very concise help and valuable help by the reviewers, in the presentation of the manuscript.

I have 2 minor additions which I suggest be added to the manuscript.

Line 95: What is your understanding/theoretical stand point in relation to complex pain, you state which approach is needed but not how you define and understand complex pain. This could be clarified.

95: “Management of chronic pain is complex, requiring a person-centred assessment [1]and multimodal approach [3].”

Clinical reasoning is a key component

Line 146-147 you define clinical reasoning for physiotherapists as a “context dependent way of thinking and decision making in professional practice to guide practice actions” [18]

Even though you describe an inductive approach in your framework analysis it seems unrealistic and maybe critical, that you do not have an informed starting point by having a model/theoretical understanding of clinical reasoning. This should be elaborated and clarified in the analysis section in Methods as this will influence the future analysis.

Reviewer #3: Thank you for your work. I feel that you managed to respond well to previous comments from colleague reviewers. There is no need for extra reviews after that.

7. PLOS authors have the option to publish the peer review history of their article (what does this mean?). If published, this will include your full peer review and any attached files.

Reviewer #2: No

Reviewer #3: No

---

## [Author Response · Author response to Decision Letter 1]

10 Aug 2023

Reviewer comment 

Line 95: What is your understanding/theoretical stand point in relation to complex pain, you state which approach is needed but not how you define and understand complex pain. This could be clarified.

95: “Management of chronic pain is complex, requiring a person-centred assessment [1]and multimodal approach [3].”

 Response: Within the script, we have used the term chronic pain, supported by the NICE guidelines definition. We have not used the term complex pain, as we are stating that the management of chronic pain is in itself complex (and requires person-centred assessment and multimodal approach). The focus of the study is clinical reasoning within chronic pain, as opposed to within complex pain, which may be a separate entity.

As we are not focusing on complex pain (which we are unaware if an accepted definition exists, or is an accepted term), we feel that discussing complex pain, or introducing this term to the paper adds unnecessary deviation from the focus of the study, namely chronic pain. 

Reviewer comment: Clinical reasoning is a key component

Line 146-147 you define clinical reasoning for physiotherapists as a “context dependent way of thinking and decision making in professional practice to guide practice actions” [18] 

Even though you describe an inductive approach in your framework analysis it seems unrealistic and maybe critical, that you do not have an informed starting point by having a model/theoretical understanding of clinical reasoning. This should be elaborated and clarified in the analysis section in Methods as this will influence the future analysis.

 Response: Within the analysis section of the methods section (line 256), we have highlighted that relevant clinical reasoning frameworks will inform analysis where appropriate, using an inductive approach to allow themes to emerge. 

As different clinicians may utilise different clinical reasoning models (such as hypothetico-deductive, pattern recognition, biopsychosocial etc), we do not want to have a specific reasoning model as a starting point of analysis.

---

## [Decision Letter · Decision Letter 2]

21 Sep 2023

PONE-D-23-00756R2Physiotherapists clinical reasoning to prescribe exercise for patients with chronic pain: a qualitative study research protocol.PLOS ONE

Dear Dr. Kelly,

Thank you for submitting your  revised manuscript to PLOS ONE. After careful consideration, we feel that it has fully meet PLOS ONE’s publication criteria as it currently stands. Therefore, we invite you to submit a revised version of the manuscript that addresses the points raised during the review process.

Kindly modify your abstract as previous into a structured abstract as follows with the following sub-headings, as recommended by the peer-reviewer:

1. Background (here include the purpose /aim of the study with a brief background).

2. Methods (summary of methods)

3. Results (summary of results/expected outcomes)

4. Conclusion.

Also address any other observations or comments by the peer- reviewers.

We look forward to receiving your revised manuscript.

Kind regards,

Sylvester Chidi Chima, M.D., L.L.M, LLD.

Academic Editor

PLOS ONE

Journal Requirements:

Reviewers' comments:

Reviewer's Responses to Questions

**Comments to the Author**

1. Does the manuscript provide a valid rationale for the proposed study, with clearly identified and justified research questions?

Reviewer #1: Yes

Reviewer #2: Yes

2. Is the protocol technically sound and planned in a manner that will lead to a meaningful outcome and allow testing the stated hypotheses?

Reviewer #1: Yes

Reviewer #2: Yes

3. Is the methodology feasible and described in sufficient detail to allow the work to be replicable?

Reviewer #1: Yes

Reviewer #2: Yes

4. Have the authors described where all data underlying the findings will be made available when the study is complete?

Reviewer #1: Yes

Reviewer #2: Yes

5. Is the manuscript presented in an intelligible fashion and written in standard English?

Reviewer #1: No

Reviewer #2: Yes

6. Review Comments to the Author

You may also provide optional suggestions and comments to authors that they might find helpful in planning their study.

Reviewer #1: The authors should re organize the abstract to be logical with the following sub headings:

1. Background (which should also include the purpose /aim of the study).I advise they cut down and provide just a summary of the background.

2. Methods (summary of methods)

3. Results (summary of results)

4. Conclusion.

There is no need to provide a discussion in the abstract.

Reviewer #2: I can accept the arguments provided by the authors No further comments. Good luck with the coming project.

7. PLOS authors have the option to publish the peer review history of their article (what does this mean?). If published, this will include your full peer review and any attached files.

Reviewer #1: No

Reviewer #2: No

---

## [Author Response · Author response to Decision Letter 2]

3 Nov 2023

Kindly modify your abstract as previous into a structured abstract as follows with the following sub-headings, as recommended by the peer-reviewer:

1. Background (here include the purpose /aim of the study with a brief background).

2. Methods (summary of methods)

3. Results (summary of results/expected outcomes)

4. Conclusion 

1) We have added the aim of the study into the background. 

2) The method has remained the same

3) As this is a protocol, we do not have any results. Therefore, we have included here what the outcome will be with the results. 

4) We have removed the subtitle discussion and replaced with conclusion, and provided a conclusion.

---

## [Decision Letter · Decision Letter 3]

22 Nov 2023

Physiotherapists clinical reasoning to prescribe exercise for patients with chronic pain: a qualitative study research protocol.

PONE-D-23-00756R3

Dear Dr. Kelly,

We’re pleased to inform you that your manuscript has been judged scientifically suitable for publication and will be formally accepted for publication once it meets all outstanding technical requirements.

Kind regards,

Sylvester Chidi Chima, M.D., L.L.M, LLD.

Academic Editor

PLOS ONE

Additional Editor Comments (optional):

We have received 3 Accept recommendations from previous peer reviewers for this study protocol. The fourth review appears to duplication of previous reports and may be considered by the authors during their full study

Reviewers' comments:

Reviewer's Responses to Questions

**Comments to the Author**

1. Does the manuscript provide a valid rationale for the proposed study, with clearly identified and justified research questions?

Reviewer #1: Yes

Reviewer #4: Partly

2. Is the protocol technically sound and planned in a manner that will lead to a meaningful outcome and allow testing the stated hypotheses?

Reviewer #1: Yes

Reviewer #4: Partly

3. Is the methodology feasible and described in sufficient detail to allow the work to be replicable?

Reviewer #1: Yes

Reviewer #4: No

4. Have the authors described where all data underlying the findings will be made available when the study is complete?

Reviewer #1: Yes

Reviewer #4: Yes

5. Is the manuscript presented in an intelligible fashion and written in standard English?

Reviewer #1: Yes

Reviewer #4: Yes

6. Review Comments to the Author

You may also provide optional suggestions and comments to authors that they might find helpful in planning their study.

Reviewer #1: Authors of the manuscript have appropriately addressed all the comments. I have no further comments.

Reviewer #4: Introduction

I understand the work claims to be the first study (line 274) seeking to under standing clinical reasoning processes of physiotherapists when describing exercise. When such a claim is made I think it is important that other articles around experience of PTs and clinical reasoning article more generally are acknowledged to give a context and a better focus on the gap so for the introduction and discussion I have had a quick search of articles (I am not an expert in this area so forgive me if these are not bang on) but I would think some of them need to be acknowledge so the context of this statement can be fully understood and acknowledgement to past work and the gaps are identified.

Frontiers | The Needs and Experiences of Patients on Pain Education and the Clinical Reasoning of Physical Therapists Regarding Cancer-Related Pain. A Qualitative Study (frontiersin.org)

Exercise prescription for patients with non-specific chronic low back pain: a qualitative exploration of decision making in physiotherapy practice - ScienceDirect

Full article: Clinical reasoning and critical reflection in physiotherapists’ examinations of patients with low back pain in its early phase: a qualitative study from physiotherapists’ point of view (tandfonline.com)

The clinical reasoning of pain by experienced musculoskeletal physiotherapists - ScienceDirect

Articles that contain some information

Physiotherapists’ beliefs and attitudes influence clinical practice in chronic low back pain: a systematic review of quantitative and qualitative studies - ScienceDirect - this review cites 5 past articles

Aspects influencing clinical reasoning and decision-making when matching treatment to patients with low back pain in primary healthcare - ScienceDirect

Self‐management and chronic low back pain: a qualitative study - Crowe - 2010 - Journal of Advanced Nursing - Wiley Online Library

Self-management support for people with non-specific low back pain: A qualitative survey among physiotherapists and exercise therapists - ScienceDirect

Full article: The suitability and utility of the pain and movement reasoning model for physiotherapy: A qualitative study (tandfonline.com)

Context articles

Full article: Factors influencing physical therapists’ clinical reasoning: qualitative systematic review and meta-synthesis (tandfonline.com)

Full article: Patients with chronic pain may need extra support when prescribed physical activity in primary care: a qualitative study (tandfonline.com)

Promoting Participation in Physical Activity and Exercise Among People Living with Chronic Pain: A Qualitative Study of Strategies Used by People with Pain and Their Recommendations for Health Care Providers | Pain Medicine | Oxford Academic (oup.com)

Methods

Lines 169-177

You call the research you are planning a generic qualitative study. The reference you use states this:

One research approach that falls under this broad category is known as generic qualitative research, which is subsequently subdivided into genres of interpretive description and descriptive qualitative research (Caelli et al., 2003).

Later on the author give a perspective on a philosophical stance – e.g.,

generic studies seek to understand how people interpret, construct, or make meaning from their world and their experiences. Furthermore, she writes that generic studies are epistemologically social constructivist, theoretically interpretive studies that focus on “(1) how people interpret their experiences, (2) how they construct their worlds, and (3) what meaning they attribute to their experiences” (Merriam, 2009, p. 23).

Given the above do you need to be more specific as to whether you work will be a descriptive or interpretive description approach? If so, there may be a better reference to use also do you need to mention something about social constructivism?

You mention interviews here but this is methods so can go later on?

Do you want to separate out aspects of setting, context and reflexivity?

Lines 178-184

The development of the guide could be expanded and include if you will do pilot interviews?

7. PLOS authors have the option to publish the peer review history of their article (what does this mean?). If published, this will include your full peer review and any attached files.

Reviewer #1: No

Reviewer #4: No

---

## [Editor Report · Acceptance letter]

24 Nov 2023

PONE-D-23-00756R3 

Physiotherapists clinical reasoning to prescribe exercise for patients with chronic pain: a qualitative study research protocol. 

Dear Dr. Kelly:

I'm pleased to inform you that your manuscript has been deemed suitable for publication in PLOS ONE. Congratulations! Your manuscript is now with our production department. 

Kind regards, 

on behalf of

Professor Sylvester Chidi Chima 

Academic Editor

PLOS ONE